# A Comparative Study of the Electroneurographic Findings in Amyloidotic Polyneuropathy in Patients with Light-Chain Amyloidosis and Glu54Gln Transthyretin Amyloidosis

**DOI:** 10.3390/medicina60122027

**Published:** 2024-12-09

**Authors:** Mirela Drăghici, Sorina N. Bădeliță, Andreea Jercan, Oana Obrișcă, Crisanda Vîlciu, Monica Popescu, Adina Turcu-Stiolica, Daniel Coriu

**Affiliations:** 1Fundeni Clinical Institute, 022328 Bucharest, Romania; sorinabadelita@gmail.com (S.N.B.); je.andreea@gmail.com (A.J.); oanaobrisca@gmail.com (O.O.); crisandavalciu@yahoo.com (C.V.); monica2982003@yahoo.com (M.P.); daniel_coriu@yahoo.com (D.C.); 2Hematology Department, University of Medicine and Pharmacy Carol Davila, 020021 Bucharest, Romania; 3Neurology Department, University of Medicine and Pharmacy Carol Davila, 020021 Bucharest, Romania; 4Biostatistics Department, Faculty of Pharmacy, University of Medicine and Pharmacy Craiova, 200349 Craiova, Romania; adina.turcu@gmail.com

**Keywords:** primary amyloidosis, hereditary transthyretin amyloidosis, electrodiagnostic studies

## Abstract

*Background and Objectives*: Amyloidosis is a disorder characterized by the abnormal folding of proteins, forming insoluble fibrils that accumulate in tissues and organs. This accumulation disrupts normal tissue architecture and organ function, often with serious consequences, including death if left untreated. Light-chain amyloidosis (AL) and hereditary transthyretin-type amyloidosis (hATTR) are two of the most common types. In amyloidosis, peripheral nervous system involvement is a significant diagnostic feature, particularly when it manifests as polyneuropathy, carpal tunnel syndrome (CTS), and dysautonomia. These neurological symptoms often point to the involvement of amyloid deposits in the peripheral and autonomic nervous systems, which can help identify and differentiate between the various types of amyloidosis. *Materials and Methods*: This retrospective study focused on the evolution of electrophysiological parameters in two groups: AL (*n* = 22) and hATTR-Glu54Gln patients (*n* = 14), with mixed axonal polyneuropathy. Patients were followed for two consecutive years to assess disease progression. The PND scale (polyneuropathy disability) was also used to assess motor impairment for each patient. *Results*: In our study AL amyloidosis patients presented with mixed, axonal polyneuropathy associated with CTS in 63.6% of cases and cardiomyopathy (45.5%). Serial EMGs (electromyography) showed decreased motor amplitudes of the common peroneal and tibial nerves and sensory amplitude of the superficial peroneal nerve, with mostly preserved conduction velocities. The patients maintained stage I PND throughout the monitoring period. The entire hATTR group displayed mixed, axonal polyneuropathy and cardiomyopathy; 85.7% of them had CTS, and 42.9% had orthostatic hypotension. EMG data showed decreased motor amplitudes of the tibial and common peroneal nerves, decreased sensory amplitudes of the superficial peroneal nerve, and mildly reduced conduction velocities, with significant progression at 12 and 24 months. The patients displayed additional reduced muscle strength, some reaching stage 3A and 3B-PND at the end of the study. *Conclusions*: The amyloidotic polyneuropathy found in the groups was similar in its axonal, sensory-motor, and length-dependent characteristics, but the study showed significant differences in its progression, with more abrupt changes in the hATTR-Glu54Gln group. The amyloidosis AL patients remained in stage 1 PND, while the hATTR-Glu54Gln patients progressed to stage 3 PND at 24 months.

## 1. Introduction

Systemic amyloidosis is a rare group of acquired or inherited diseases, that is based on the extracellular storage of abnormal proteins, leading to multiple organ damage and complex symptoms. The most common types of amyloidosis are light-chain amyloidosis (AL) and hereditary amyloidosis with mutations in the transthyretin gene (hATTR), with various phenotypes according to the mutation identified and age of onset [1]. In both cases, peripheral nervous system (PNS) damage begins in the early stages, with the patient frequently experiencing painful paresthesia in the limbs [2]. Also, isolated carpal tunnel syndrome (CTS) may predate the diagnosis of amyloidosis for several years [3,4]. However, the main negative prognostic factor with a direct impact on mortality is amyloidotic cardiac infiltration [5]. Therefore, detailed investigations to determine the etiology of a mixed axonal polyneuropathy associated with carpal tunnel syndrome and dysautonomia may lead to an early diagnosis of amyloidosis, with therapeutic and prognostic implications.

AL amyloidosis is a systemic, malignant condition caused by medullary plasma cells’ production of light chains of immunoglobulins, which are deposited systemically [1,6]. The frequency of neuropathy in patients with AL amyloidosis ranges from 9.6 to 35%, initially involving thin nerve fibers, later progressing to mixed axonal polyneuropathy and vegetative neuropathy [7,8,9].

Hereditary transthyretin-type amyloidosis (hATTR) is a rare, autosomal-dominant, adult-onset disease caused by extracellular deposition of conformationally altered transthyretin produced mainly in the liver. The peripheral nervous system and heart are the main tissues affected, leading to axonal sensorimotor polyneuropathy and amyloidotic cardiomyopathy [9]. In endemic countries (e.g. Portugal, Brazil, Japan, and Sweden) this disease is easier to recognize and diagnose, especially when it is associated with the classic phenotype, characterized by sensory-motor polyneuropathy associated with significant damage to thin fibers and dysautonomia in the context of a positive family history and increased penetrance of the mutation [10,11]. More than 130 mutations are responsible for the occurrence of hATTR, associating complex phenotypes, ranging from predominantly peripheral nerve damage to cardiac or mixed phenotypes [12,13]. The evolution of hATTR varies, but it is an aggressive disease with a survival time from symptom onset between 6 and 12 months, with a negative prognosis mainly due to amyloidotic cardiomyopathy and its complications [11].

## 2. Materials and Methods

### 2.1. Study Design and Participants

We performed a retrospective study of patients with AL and hATTR amyloidosis diagnosed at the Fundeni Clinical Institute, Hematology Clinic in Bucharest, Romania. This study aimed to compare electrodiagnostic parameters between two cohorts of patients with AL amyloidosis and symptomatic hATTR variant Glu54Gln diagnosed at the Fundeni Hematology Clinic between 2000 and 2021.

The inclusion criteria were patients with AL amyloidosis and neurological impairment and patients with symptomatic hATTR Glu45Gln variant, with three consecutive electrodiagnostic studies at enrollment, 12 months, and 24 months. We excluded carriers of the Glu54Gln mutation from the families of hATTR patients.

Demographic data (age, sex), year of diagnosis, onset-diagnosis duration, presence of neurological and cardiological impairment, and electromyography (EMG) data were collected from the patient’s electronic medical records and clinical interviews.

The neurological involvement was concluded by polyneuropathy, confirmed by EMG investigation. Regarding cardiac involvement, patients underwent thorough cardiac evaluation, and the diagnosis of amyloid cardiomyopathy, based on Gertz et al., 2005 definitions [14].

### 2.2. Neurological Assessment

Electromyographic studies were performed on all patients, by stimulating the common peroneal and bilateral tibial nerves, at different intensities: 40, 50, or max. 100 mA.

### 2.3. Hematological Assessment

Complete blood count, serum protein electrophoresis, serum cardiac markers, and serum-free light-chain assay were performed on all patients.

### 2.4. Cardiological Assessment

Echocardiography was performed on all patients.

### 2.5. Statistical Analysis

A descriptive analysis was conducted to examine the sociodemographic and clinical variables. Continuous variables are reported as median and interquartile range, along with mean and standard deviation. Frequencies and percentages were calculated for the qualitative variables. The Related Samples Wilcoxon Signed-Rank test was used to compare the electrophysiological measurements among the included patients for three consecutive years. Spearman’s correlation coefficients were computed to determine the associations between variables. All statistical analyses were performed using GraphPad Prism v.9.5.1 for Windows (GraphPad Software, San Diego, CA, USA; www.graphpad.com, accessed on 2 October 2024). The significance level was set as *p* < 0.05.

## 3. Results

A total of 223 patients were enrolled in the 2000–2021 interval, 195 patients with AL, and 28 patients with symptomatic hATTR-Glu54Gln. The mean age of the entire group was 60 years, with a significant difference between the AL (62 years) and hATTR-Glu54Gln (44.6 years) groups (*p* = 0.000, *t*-test). There were no statistically significant differences in the male/female ratio between the two groups (1.09 for AL and 0.67, hATTR-Glu54Gln, *p* = 0.076). The duration from disease onset to diagnosis was significantly longer for patients with ATTRh-Glu54Gln than for those with AL (27 months vs. 12.5 months, *p* = 0.000) (Table 1).

Fourteen hATTR patients and 22 with AL amyloidosis had three consecutive EMG studies performed. The age of patients with hATTR was found to be significantly lower (48 years) than that of patients with AL amyloidosis (57.3 years) (*p* = 0.002). No significant differences were observed between the two patient groups in terms of sex or certain clinical variables (CTS and orthostatic), as shown in Table 1. However, cardiac involvement was significantly more prevalent in hATTR patients (100%) than in AL amyloidosis patients (45.5%) (*p* = 0.001). Furthermore, at baseline, constipation was found to be significantly more common among hATTR (64.3%) patients than among AL amyloidosis (27.3%) patients (*p* = 0.041).

The results of the electrophysiological measurements of AL amyloidosis patients are outlined in Table 2. Significant differences were observed in the decline of electrophysiological measurements between the first and second evaluations, as well as between the 12-month and 24-month evaluations.

The compound muscle action potential (CMAP) of the common peroneal nerve showed a non-significant decrease at the 12-month evaluation (*p* = 0.280). However, there was a significant decrease in CMAP at the 24-month evaluation (*p* = 0.013). Notably, in Figure 1, we observed increased CMAP values for the common peroneal nerve in certain patients at 12 months (patients no. 3–8, 12, 13, and 17).

The results of electrophysiological measurements of the hATTR-Glu54Gln patients are presented in Table 3. Significant differences were observed in the decrease in nearly all electrophysiological measurements at the 12-month evaluation, as well as the 24-month evaluation. However, a significant decrease was found in the motor conduction velocity (MCV) of the common peroneal nerve at the 12-month EMG exam (*p* = 0.041), and no significant decrease was found at 24 months (*p* = 0.272). Although an increase in the CMAP value was observed in the common peroneal nerve at 24 months, it was not statistically significant (*p* = 0.157).

As depicted in Figure 2, we observed severe MCV slowing of the tibial nerve, reaching 0 m/s in four hATTR patients at 24 months. The mean ± SD decrease in CMAP of the tibial nerve was 36 ± 33% at 12 months and 46 ± 43% at 24 months in hATTR-Glu54Gln patients. Conversely, the decrease was slower in patients with AL amyloidosis: 6 ± 39% at 12 months and −1 ± 74% at 24 months.

For the common peroneal nerve, the mean ± SD decrease in CMAP was 32 ± 27% at 12 months and 6 ± 49% at 24 months in hATTR-Glu54Gln patients, whereas it was slower in AL amyloidosis patients: −1 ± 22% at 12 months and 5 ± 13% at 24 months.

In terms of the MCV of the tibial nerve in the hATTR-Glu54Gln group, the mean ± SD decrease was 12 ± 27% at 12 months and 31 ± 43%, whereas it was slower in the AL amyloidosis group (7 ± 6% at 12 months and 4 ± 7% at 24 months). Lastly, for the common peroneal nerve, the mean ± SD decrease in MCV was 11 ± 17% at 12 months and 9 ± 14% at the 24-month EMG evaluation in hATTR-Glu54Gln patients, whereas it was slower in AL amyloidosis patients: 4 ± 8% at 12 months and 6 ± 9% at 24 months.

Regarding the sensory nerve action potential (SNAP) in the superficial peroneal nerve, the mean ± SD decrease was 27 ± 34% at 12 months and 35 ± 40% at 24 months in patients with hATTR amyloidosis, whereas it was slower in patients with AL amyloidosis (21 ± 22% at 12 months and 12 ± 18% at 24 months). In addition, the mean ± SD decrease in sensory conduction velocity (SCV) slowing of the superficial peroneal nerve was 5 ± 10% at the 12-month evaluation and 27 ± 43% at 24 months in the hATTR-Glu54Gln group, whereas it was slower in AL amyloidosis group (7 ± 12% at 12 months and 4 ± 8% at 24 months).

Additionally, we investigated whether the decline in electrophysiological measurements at 12 and 24 months correlated with disease stage. No significant correlations were observed, except for the MCV of the tibial nerve at 12 months in patients with AL amyloidosis (rho = −0.459, *p*-value = 0.036). This indicates that the conduction velocity of the tibial nerve exhibited a greater decrease in patients with AL amyloidosis in the initial stage.

In the AL amyloidosis group, 19 individuals (90.48%) were initially diagnosed with stage 1 PND, whereas two patients (9.52%) were diagnosed with stage 2 PND. The disease stages remained unchanged during the follow-up period and the patients were under treatment. All patients with AL amyloidosis received subcutaneous bortezomib-based therapeutic regimens, usually in combination with cyclophosphamide and dexamethasone.

During the initial evaluation of the hATTR-Glu54Gln patients, nine of them (69.23%) were in stage 1 PND and four (30.77%) were in stage 2. At the end of the follow-up period, only three patients (23.08%) in stage 1 maintained their baseline stage. The remaining patients exhibited the following changes: five patients (38.46%) progressed to stage 2, one patient (7.69%) advanced to stage 3A, three patients (23.08%) reached stage 3B, and one patient (7.69%) reached stage 4.

## 4. Discussion

The present study focuses on the progression of two of the most common forms of amyloidotic polyneuropathies, AL and hATTR Glu54Gln amyloidosis, by observing 36 subjects over a 3-year study period. The Glu54Gln mutation is the most common hATTR mutation encountered in Romania and is associated with a mixed phenotype manifested by severe axonal sensorimotor polyneuropathy, dysautonomia, and significant cardiac impairment which is the main factor contributing to the short survival of patients [15].

The AL amyloidosis cohort was introduced because these two neuropathies have many similarities at disease onset, but their evolution is quite variable and different. By performing serial electrodiagnostic studies in these patients, we aimed to identify significant changes in the electrophysiological parameters that reflect the pattern of progression.

Nerve conduction studies for the two groups displayed classical findings suggestive of axonal sensorimotor neuropathy, similar to the literature data on amyloid neuropathies [6]. However, demyelinating features have been encountered in other ATTRh cohorts, as over a third of them might fulfill the criteria for chronic demyelinating polyneuropathy, as shown by Davion et al. [16]. Regarding our group of patients with the hATTR—Glu54Gln variant, there is a homogenous involvement of the PNS expressed mainly through axonal, sensory-motor polyneuropathy, accompanied in a few cases by secondary slowing of the motor conduction velocities but without being in a demyelinating range.

In electrophysiological studies of AL amyloidosis, CMAP of the common peroneal nerve showed a statistically significant decrease at 24 months. A significant reduction in the amplitude of the superficial peroneal nerve was also observed at 24 months. Both motor and sensory conduction velocities remained within normal limits at 12 months and decreased slightly but not significantly at 24 months, demonstrating a reduction in nerve fibers, secondary to axonal loss.

Therapeutic regimens applied to AL amyloidosis patients contained Bortezomib, which is known for its neurotoxicity [17]. In patients with AL amyloidosis and neurological impairment at diagnosis, we can discuss possible additional neurotoxicity and its involvement in worsening electrophysiological parameters. In patients with severe neurological impairment at diagnosis, replacement of bortezomib from the therapeutic regimen may be considered.

In hATTR, the conduction study showed significant decreases in all electrophysiological nerve parameters at both 12 and 24 months of follow-up, but a significant decrease in the MCV of the common peroneal nerve was observed at 12 months. In four patients with ATTRh, the MCV of the common peroneal nerve decreased severely, reaching 0 m/s at 24 months of follow-up. MCV of the tibial nerve showed a greater decrease in patients with AL amyloidosis at a 12-month follow-up.

The superficial peroneal sensory nerve showed a slow decrease in SNAP of 35 ± 40% at 24 months follow-up in AL amyloidosis patients compared to TTR patients, where the decrease was even slower, 21 ± 22% for the same period.

In the AL amyloidosis group, there was a trend of correlation between the decrease in CMAP amplitude for the tibial and common peroneal motor nerves with PND staging, more so at the 12- and 24-month follow-ups, but not statistically significant, with patients reaching at most stage I on the PND scale.

Moreover, for hATTR patients, a trend of correlation of the reduction in the tibial and fibular head CMAP amplitude and the degree of motor severity was observed as early as 12 months, but especially at 24 months of follow-up, when patients reached stage 3B of the PND scale, with statistically insignificant differences.

However, our study has several limitations, such as the relatively small number of patients included in both groups, the limited PND scale used to correlate the EMG findings with the clinical picture, and the lack of data regarding other neurological symptoms. Furthermore, certain pain scales may have been useful in assessing painful sensory symptoms that are frequently encountered in patients with amyloidotic neuropathies.

## 5. Limitations

Worldwide, over 120 mutations in the TTR gene have been described, which cause familial transthyretin amyloidosis. In Romania, the dominant mutation is Glu54Gln, found in the northeastern region, but other mutations such as Val30Met, Glu90Lys, Glu89Val, and Ile107Val have also been described. Our study is limited to patients with the Glu54Gln mutation.

## 6. Conclusions

Peripheral nerve damage in patients with AL and hATTR amyloidosis appears to be more severe than the clinical manifestations would indicate. Although nerve parameters examined in AL amyloidosis patients decreased over the last 24 months, they remained in stage I PND.

On the other hand, electrophysiological studies of the hATTR group showed a significant impairment of nerve parameters starting with the initial evaluation, with patients in the first and second PND stages. Subsequently, at 24 months of follow-up, some patients showed no motor or sensory response on EMG, although clinically, they were in stage 3A or 3B PND.

We observed a significant decrease in motor and sensory nerve parameters at 24 months in AL amyloidosis patients compared to the hATTR group, where the decrease was present and severe since 12 months. The amplitudes and conduction velocities of the monitored nerves were severely and early reduced in the hATTR group compared to those in the AL amyloidosis group, where they decreased significantly at the end of monitoring.

In patients with AL amyloidosis and neurological impairment at diagnosis, possible additional therapy-related neurotoxicity and its involvement in worsening electrophysiological parameters can be discussed. In patients with severe neurological impairment at diagnosis, replacement of bortezomib from the therapeutic regimen should be considered.

## Figures and Tables

**Figure 1 medicina-60-02027-f001:**
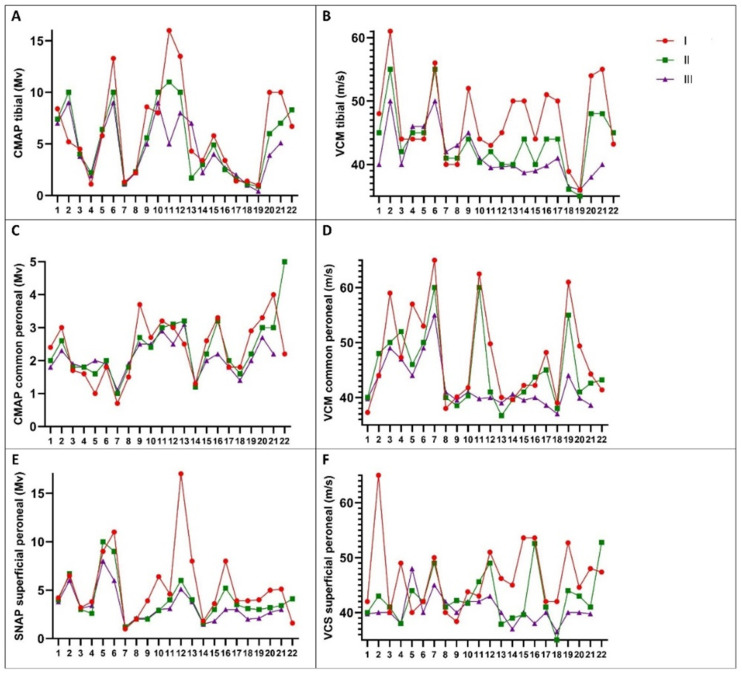
Electrophysiological data of AL patients. (**A**), CMAP tibial nerve. (**B**), MCV tibial nerve. (**C**), CMAP common peroneal nerve. (**D**), MCV common peroneal nerve. (**E**), SNAP superficial peroneal nerve. (**F**), SCV superficial peroneal nerve.

**Figure 2 medicina-60-02027-f002:**
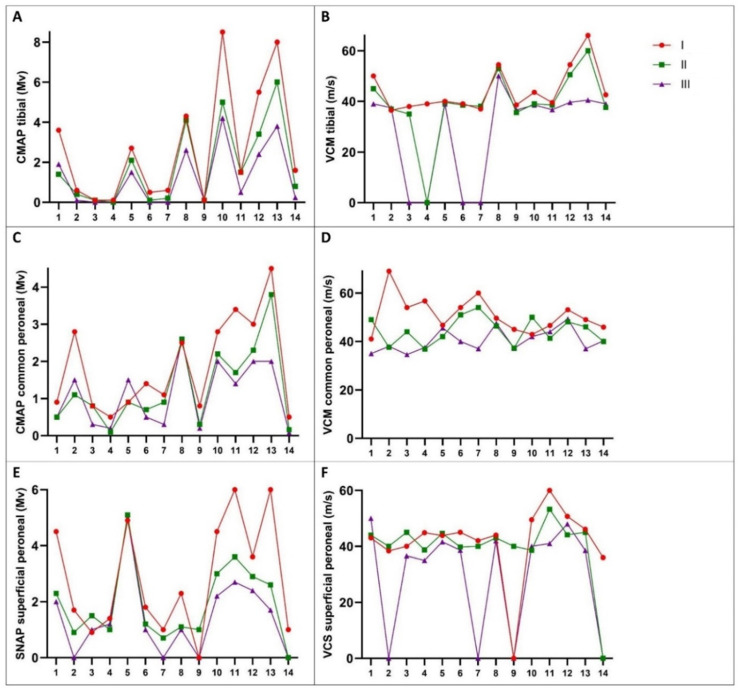
Electrophysiological measurements for hATTR-Glu54Gln patients. (**A**), CMAP of the tibial nerve. (**B**), MCV of the tibial nerve. (**C**), CMAP of the common peroneal nerve. (**D**), MCV of the common peroneal nerve. (**E**), SNAP of the superficial peroneal nerve. (**F**), SCV of the superficial peroneal nerve.

**Table 1 medicina-60-02027-t001:** Demographics and clinical characteristics of patients.

Characteristics	hATTR Amyloidosis (*n* = 14)	AL Amyloidosis (*n* = 22)	*p*
Age at time of diagnosis (years)	48.00 ± 8.86 48.5 (41.75–52)	57.27 ± 7.51 57 (51.75–64)	0.002 ^a^
Gender, Male, *n* (%)	2 (14.3%)	10 (45.5%)	0.076 ^b^
Cardiac involvement, yes, *n* (%)	14 (100%)	10 (45.5%)	0.001 ^b^
STC, yes, *n* (%)	12 (85.7%)	14 (63.6%)	0.255 ^b^
Orthostatic, yes, *n* (%)	6 (42.9%)	9 (40.9%)	1.000 ^b^
Constipation, yes, *n* (%)	9 (64.3%)	6 (27.3%)	0.041 ^b^

^a^ Mann–Whitney U tests; ^b^ Fisher’s Exact test.

**Table 2 medicina-60-02027-t002:** Comparison of electrophysiological data among AL amyloidosis patients for three consecutive years.

AL Patients (*n* = 22)		At Diagnosis (I)	At 12 Months (II)	At 24 Months (III)	*p*-Value ^a^ I vs. II II vs. III
Tibial nerve	CMAP (Mv)	6.15 ± 4.4 5.5 (2.0–8.95)	5.32 ± 3.47 5.3 (2.1–8.7)	4.54 ± 2.8 4 (2.1–7)	0.088 0.004
	VCM (m/s)	46.96 ± 6.26 44.5 (43.2–51.3)	43.6 ± 4.9 44 (40.23–45)	41.47 ± 3.89 40 (39.25–44)	0.001 0.036
Common peroneal nerve	CMAP (Mv)	2.36 ± 0.9 2.5 (1.68–3.1)	2.38 ± 0.87 2.2 (1.8–3)	2.09 ± 0.5 2 (1.8–2.5)	0. 515 0.013
	VCM (m/s)	47.37 ± 8.63 44.2 (40.1–54)	45.07 ± 6.83 42.9 (40–50)	42.2 ± 4.47 40 (39.5–44.0)	0.044 0.004
Superficial peroneal nerve	SNAP (Mv)	5.34 ± 3.62 4.1 (3.5–6.9)	3.98 ± 2.22 3.3 (2.83–4.38)	3.32 ± 1.69 3 (2.1–3.8)	0.004 0.002
	VCS (m/s)	46.33 ± 6.3 44.8 (42–50.3)	42.84 ± 4.57 41.9 (39.9–44.4)	40.53 ± 2.58 40 (39.8–42)	0.027 0.027

Continuous variables are presented as mean ± standard deviation and median (interquartile range); ^a^, Related Samples Wilcoxon Signed-Rank test.

**Table 3 medicina-60-02027-t003:** Comparison of the electrophysiological data of hATTR-Glu54Gln patients for three consecutive EMG studies.

hATTR-Glu54Gln Patients (*n* = 14)		At Enrollment (I)	At 12 Months (II)	At 24 Months (III)	*p* ^a^ I vs. II II vs. IIII vs. II
Tibial nerve	CMAP (Mv)	2.69 ± 2.9 1.6 (0.4–4.6)	1.8 ± 2.03 1.1 (0.1–3.58)	1.24 ± 1.5 0.37 (0–2.5)	0.003 0.008
	MCV (m/s)	44.2 ± 8.76 39.8 (38.5–51.13)	39.211 ± 13.48 38.6 (36.7–46.38)	28.3 ± 18.86 38.1 (0–39.15)	0.005 0.016
Common peroneal nerve	CMAP (Mv)	1.85 ± 1.28 1.25 (0.8–2.9)	1.29 ± 1.08 0.9 (0.45–2.23)	1.08 ± 0.87 0.95 (0.3–2)	0.003 0.157
	MCV (m/s)	50.96 ± 7.5 49.3 (45.7–54.7)	44.54 ± 5.52 45 (39.4–49.3)	40.36 ± 4.66 39 (37–44.4)	0.041 0.272
Superficial peroneal nerve	SNAP (Mv)	2.83 ± 2.03 2.1 (1.0–4.6)	1.92 ± 1.39 1.35 (0.98–2.93)	1.44 ± 1.39 1.1 (0–2.3)	0.016 0.003
	SCV (m/s)	41.7 ± 13.34 44 (39.6–47)	39.72 ± 12.07 41.5 (39.5–44.7)	29.38 ± 19.68 38.6 (0–41.7)	0.149 0.039

Continuous variables are presented as mean ± standard deviation and median (interquartile range); ^a^, Related Samples Wilcoxon Signed-Rank test.

## Data Availability

All data reported within the article are available anonymously from qualified investigators upon request.

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
