# Peer review of "A Comparative Study of the Electroneurographic Findings in Amyloidotic Polyneuropathy in Patients with Light-Chain Amyloidosis and Glu54Gln Transthyretin Amyloidosis"

_medicina, 2024, doi:10.3390/medicina60122027_

Round 1
Reviewer 1 Report
Comments and Suggestions for Authors
Thank you for the opportunity to review your manuscript titled: A comparative study of the electroneurographic findings in amyloidotic polyneuropathy in patients with light chain amyloidosis and Glu54Gln transthyretin amyloidosis
The authors present a valuable, more in depth sequel to their original publication on the clinical characteristics of patients with hereditary amyloidosis associated with the Glu54Gln pathogenic variant in transthyretin, comparing these findings to those observed in AL amyloidosis.
Please see below for the comments/suggestions.
· Please consider replacing word “mutation” with “pathogenic variant” in accordance with the ACMG classification criteria.
· Authors may consider adding more information regarding the variant-: This sequence change replaces glutamic acid, which is acidic and polar, with glycine, which is neutral and non-polar, at codon 74 of the TTR protein (p.Glu74Gly). This variant is also known as Glu54Gly. This missense variant has been observed in individuals with transthyretin (TTR) amyloidosis and related cardiomyopathy
· Line 70- 71 The evolution of hATTR varies, but it is an aggressive disease with a survival time from symptom onset between 6 and 12 months, with a negative prognosis mainly due to amyloidotic cardiomyopathy and its complications. [11]
With the advent of early screening and improved diagnostic methods, this reference (from 2002) may now be outdated.
· Line 83 - We excluded carriers of the Glu54Gln mutation from the families of hATTR patients-
The term "carrier" is typically used in the context of autosomal recessive conditions. An individual who carries a single pathogenic or likely pathogenic genetic variant in a gene associated with a recessive condition may be referred to as a carrier. However, amyloidosis is an autosomal dominant condition, meaning that anyone with a pathogenic variant in TTR, such as Glu54Gln, who remains asymptomatic, may still carry the genetic mutation but show no overt signs of the disease. The question then arises: are these individuals truly asymptomatic, or do they have very subtle symptoms that have yet to be diagnosed?
It may be interesting to consider this population as a third group in this study, as little is currently known about them. Genetic testing has only recently enabled the identification of such individuals, and understanding any subtle symptoms they might experience could provide valuable insights into the early stages of the disease
· Line 90 – ACC expert consensus recommends the use of 99m Tc PYP to diagnose cardiac amyloidosis caused by ATTR proteins. Was this performed? If not, were other causes of cardiomyopathy excluded?
· Line 116- age at diagnosis 44- was this age at diagnosis different because of the availability of early genetic screening? Usual age at diagnosis for hATTR is later.
The duration from disease onset to diagnosis was significantly longer for patients with ATTRh-Glu54Gln than for those with AL (27 months vs12.5 months, p = 0.000).
The authors may consider elaborating on possible causes for these differences.
· Line 125 . “However, cardiac involvement was significantly more prevalent in hATTR patients (100%) than in AL amyloidosis patients (45.5%) (p = 0.001)”. Authors mention cardiac evaluation of the patient but discussion regarding comparison of cardiac markers is missing. I understand that the title specifies the electroneurographic findings in these two group, however, please consider expanding on this topic for further clarity in the discussion section and conclusion.
· Lane 185 “The disease stages remained unchanged during the follow-up period and the patients were under treatment”- for patients with AL diagnosis. Could the authors commend on whether hATTR-Glu54Gln patients were under treatment and what is their therapeutic recommendation for this group?
Author Response
1.
I sincerely appreciate your insightful suggestion to replace the term mutation with pathogenic variant in accordance with ACMG guidelines. Your input has not only improved the clarity and accuracy of my work but also ensured its alignment with current best practices in the field.
2.
Thank you for your feedback on my manuscript. Regarding the reference from 2002 on the diagnosis of TTR amyloidosis, I would like to note that this source may now be considered outdated due to advancements in early screening protocols and improved diagnostic methods. To provide a more accurate and current perspective, I have referred to the First European Consensus for Diagnosis and Treatment of Transthyretin Familial Amyloid Polyneuropathy (Portugal, 2016). This document offers an updated and comprehensive guide that aligns with modern clinical practices and reflects the latest developments in the field.
3.
Thank you for your observation regarding the use of the term carrier for the Glu54Gln mutation, which is typically associated with autosomal recessive disorders, and your suggestion to address whether these individuals are truly asymptomatic or exhibit subtle, undiagnosed symptoms. To clarify, the individuals carrying the Glu54Gln mutation were thoroughly evaluated through electrocardiographic, cardiologic, and electrophysiological examinations. In the absence of clinical symptomatology, these individuals can be considered healthy carriers of the Glu54Gln gene.
4.
I would like to confirm that, in alignment with the ACC expert consensus, we utilized 99mTc-PYP scintigraphy for all patients included in the study. This imaging method demonstrated myocardial uptake of the radiotracer, indicating the presence of amyloid deposits within the myocardial structure. This approach provided a reliable and non-invasive means to confirm the diagnosis of cardiac amyloidosis in our cohort.
5.
Thank you for your observation regarding the age of diagnosis for this mutation. While the average age of diagnosis for this mutation is indeed around 40 years, in our study, the mean age was 44 years. This slightly delayed age of diagnosis can be attributed to the low awareness of this disease within Romania, which often results in delayed recognition and diagnosis of symptoms. This highlights the need for improved education and understanding regarding this condition among healthcare professionals in the region.
6.
Thank you for your observation regarding line 125 in the article: "To provide more clarity in the Discussion and Conclusion sections", I would like to emphasize that we utilized 99mTc-PYP scintigraphy and echocardiography as exploratory methods. These imaging and cardiologic techniques are instrumental in defining clear criteria for cardiac involvement in this pathology. They played a key role in identifying and characterizing the extent of myocardial amyloid deposition in our study. I hope this explanation adds clarity, and I am happy to incorporate further refinements if needed.
7.
Regarding the absence of treatment details for the TTR group in the manuscript, I would like to clarify that the group with ATTR amyloidosis carrying the Glu54Gln mutation was treated with a transthyretin tetramer stabilizer. The therapeutic recommendation was made by the oncohematologist , prescribing Vindaquel (20 mg/day) throughout the survival period of these patients. This treatment approach aligns with current guidelines for managing transthyretin amyloidosis
Reviewer 2 Report
Comments and Suggestions for Authors
Some revisions are needed before publication

Author Response
The standardized neurological and cardiological examination ensures safety, relevance, and comparability across patients. Choosing to use a neuropathy assessment scale at this stage reflects a practical and efficient approach to prioritizing investigations, conserving resources, and tailoring treatment based on the severity of clinical symptoms.
Regarding the presentation of additional genetic aspects (pedigree studies), we fear that this is not possible because the ancestors (from the genetic tree) of some patients included in the current study have passed away.
Given the existence of a European Consensus (First European Consensus for Diagnosis, Management, and Treatment of Transthyretin Familial Amyloid Polyneuropathy, (https://pmc.ncbi.nlm.nih.gov/articles/PMC4739312/), which also serves as a guideline, we believe that our contribution, based on the results of the study we conducted, does not provide sufficient new insights to justify adding new annexes or restructuring this already well-established 2016 consensus.
Reviewer 3 Report
Comments and Suggestions for Authors
1) As this is a variant (hATTR-Glu54GLN) that is relatively predominant in Romania, more information about it should be highlighted in the introduction:
a) Founder effect?
b) Origin?
c) Epidemiology
d) Variant location in the ATTR gene
e) Main clinical manifestations
2) Did patients with the hereditary form receive any medication during medical follow-up?
3) In the future studies, I suggest carrying out dysautonomia analysis and electrophysiological studies for thin fibers.
Author Response
Thank you for your insightful comment regarding the potential founder effect of the Glu54Gln mutation in Romania. We understand the importance of exploring this aspect, as it could provide valuable information about the mutation's genetic history and epidemiology in the region. More than that,Our study focused primarily on identifying and characterizing the Glu54Gln mutation among the patients included, without conducting extensive analyses to establish the presence or absence of a founder effect. Determining a founder effect requires comprehensive genealogical and genetic studies, including haplotype analysis and ancestral tracing, which were beyond the scope of this study.
The patients in our study received treatment with a medication designed to stabilize the transthyretin tetramer. The medication used was Vyndaqel tablets, administered orally at 20 mg/day, throughout the patients' survival period.We hope this clarifies the aspect regarding the therapeutic approach used in our study.
Thank you for your valuable suggestion regarding complementary studies such as SUDOSCAN and QST. We have taken your recommendation into account and recognize that these methods could undoubtedly provide new and relevant information.
We hope to include these complementary approaches in future research, aiming to ensure consistency within the patient cohort and further expand the understanding of the studied condition.
Reviewer 4 Report
Comments and Suggestions for Authors
The present manuscript describes a multidisciplinary condition and is focused on patients with AL and hATTR amyloidosis.
The title describes well the content of the manuscript, and the abstract is well structured, informative and sending clear message to the reader, as the most important findings are presented there.
According to the introduction in my opinion is a little bit too brief, as it could give more information about the systemic amyloidosis, but yet it is an autthor's decision. There is enough information about the genetic basis of the condition, and the most common clinical presentations.
The methodology is well described as the study includes patients from a long period of time - 20years. This part presents understandably for the general reader the whole process of the investigation, the exams performed and follow up. There are no recommendations for this section.
It must be noted that 223 included patients with this pathology is a large portion for a relatively rare condition, so this improves the overall value of this research. The tables are well designed with no missing or additional information, just maybe a list of the used in the tables abbreviations would be beneficial for the better understanding when reading just the table. The rest of the section requires no further corrections.
It makes a great impression about the depth of understanding and discussing the results. Latest and most important articles in this topic were used in order to analise the results.
The limitations are noted well as they might be a small limitation for generalising the results for more patients with AL. Further research in this area involving more genetic variants or comparison with the present study would be of great interest.
The conclusions are following the main course of the work and are consistable with the present literature and give more light in the topic.
Overall, the present manuscript is of high scientific and practical value, and it is presented in a optimal manner requiring no major revisions.
Author Response
Thank you very much for your thoughtful review and for considering our article worthy of publication. We greatly appreciate your minor suggestions, which have helped us refine and improve the clarity of our work. Your positive feedback is highly encouraging, and we are grateful for your support in the publication process.